# Microstructure, Microhardness and Tribological Properties of Bronze–Steel Bimetallic Composite Produced by Vacuum Diffusion Welding

**DOI:** 10.3390/ma15041588

**Published:** 2022-02-20

**Authors:** Xiaoming Wang, Boen Tang, Linlin Wang, Dongyun Wang, Weiping Dong, Xiping Li

**Affiliations:** Key Laboratory of Urban Rail Transit Intelligent Operation and Maintenance Technology & Equipment of Zhejiang Province, Zhejiang Normal University, Jinhua 321004, China; wangxm@zjnu.cn (X.W.); tangboen2022@163.com (B.T.); zsdwdy@zjnu.cn (D.W.); dwp@zjnu.cn (W.D.); lxp2005@163.com (X.L.)

**Keywords:** tribological properties, bimetallic structure, composite, lead–bronze, vacuum diffusion welding

## Abstract

In this paper, a lead–bronze/steel bimetal composite was produced by vacuum diffusion welding technology. The microstructure, hardness and tribological properties under the dry sliding condition of the bimetal structured material were investigated and compared with two reference samples, i.e., lead–bronze and Mn/Si–brass. The wear mechanism of the three materials was also analyzed in detail. It was found that the bimetallic structure possessed the best wear resistance among the three samples. When paired with the ball bearing steel, the wear rates of the lead–bronze and Mn/Si–brass were 13 and 54 times higher than that of the bimetal composite. When paired with bearing steel, the wear rates of the two materials were 13 and 54 times higher than the bimetallic composite, respectively. This is because the steel layer served as a bearing layer to decrease the plastic deformation of the bronze layer. Furthermore, the lead can accelerate the formation of a dense hardened layer at the sliding interfaces to avoid subsequent wear of the bronze surface. Nevertheless, this hardened layer caused severe scuffing on the steel balls. Therefore, lead–bronze/steel structured material is recommended to match with hard counterface material, such as cemented carbide.

## 1. Introduction

Axial piston pumps are widely used in high-duty hydraulic fluid power systems due to their advantages of high-power densities, high-limit load pressure, and high overall efficiency [1,2,3]. One of the main friction pairs in the axial piston pumps is the valve plate/cylinder block pair. Wear of valve plate and cylinder block will cause a loss of efficiency in the axial piston pumps [4]. The valve plate and bore are contacted and slide with each other. Normally, there is an oil film between the interfaces, which plays the role of bearing and lubricating. However, when the piston pump has just started and the oil film has not yet formed, this may cause certain damage of the contact surfaces [4,5]. Meanwhile, with the current rapid development of hydraulic fields, the demand for high-pressure axial piston pumps is increasing [6]. Under high pressure, the oil film is more prone to failure [7]. At this time, the valve plate and block are in a state of dry friction or boundary friction, which can easily lead to wear damage or even burning accidents. To minimize the friction coefficient and wear between the metal pairs, efforts need to be taken on material optimization. 

Hard coatings such as TiN [8], CrSiN [9], CrZrN [9], and DLC (diamond-like carbon) [10] produced by physical vapor deposition (PVD) or chemical vapor deposition (CVD) have been investigated, and achieved good wear resistance and low friction when applied on the friction pairs in axial piston pumps. However, the high cost of these coatings limits their applications. In the work of Zhang et al. [6], it was found that ceramic materials performed better than a steel alloy (code: 30Cr2MoVA) using a bearing steel (code: GCr15) counterface under ultra-high pressure and boundary lubrication conditions. The above studies were only conducted under laboratory conditions. However, the movement forms of the friction pairs in the actual pumps are very complex, and there are both impact and sliding at the same time. Ceramic materials are prone to brittle fracture under impact. Thus, the application feasibility of ceramic materials in axial piston pumps needs to be further discussed. So far, metals are still the main application materials of key friction pairs in axial piston pumps. In the work of Jiang et al. [11] tribological tests were conducted on friction pairs between soft materials including Mn-brass, Al-bronze, nodular iron and steel-based hard materials under a dry sliding condition. It was found that the friction pairs of an Mn-bass (code: HMn58-3) and a carbonitriding-treated alloy steel (code: 20CrMnTi) had the best wear resistance.

Copper alloys are applied in axial piston pumps due to their high thermal conductivity, mechanical strength, good machinability, etc. [4,12]. Among the full range of copper alloys, leaded bronze was widely used in bearings. As a soft phase, lead is conducive to forming a lubricating film to reduce the friction and wear of bronze [13]. Thus, the lead–bronze used in the valve plate/cylinder block pair is expected to obtain good performance. However, the application of lead is restricted due to its harmful effects. Therefore, it is imperative to develop low-lead or lead-free alternative alloys. Some promising results have already been obtained when lead in brasses was substituted with bismuth [14], titanium [15] or silicon [16,17,18]. In addition, it has been reported that Mn and Si can improve the wear resistance of the alloys by forming Mn_5_Si_3_ precipitated particles [19]. Other than changing the composition of the alloys, constructing bimetal structures is proposed to be a more effective method to combine the advantages of multiple materials and improve the wear resistance of friction pairs. Of various preparation technologies of composites, dissimilar welding technologies, such as laser cladding [20,21,22], friction welding [23,24,25], diffusion welding [26,27,28,29] and vacuum diffusion welding [30] are low-cost and effective and are suitable for the industrial production. 

Among these technologies, vacuum diffusion welding has various advantages: low heating temperature, little effect on the properties of the substrate material, similarity in performance of the welded joint with respect to the substrate material, and small deformation [1]. However, research on the properties of bimetallic materials prepared by vacuum diffusion welding, especially in terms of friction properties, is still lacking.

In this paper, a lead–bronze layer with a millimeter-scale was jointed with a steel backboard by vacuum diffusion welding technology. This low-lead copper-based material acted as friction pairs of axial piston pumps operated under extreme working conditions. The tribological properties of the lead–bronze/steel composite were investigated and compared with those of two reference materials, i.e., lead–bronze and Mn/Si-modified brass. The wear mechanisms of these materials were also discussed and analyzed. 

## 2. Materials and Methods

### 2.1. Specimen Preparation

Commercial lead–bronze (ZY127401^®^) and lead-free brass with Mn and Si (ZY331604^®^) plates were obtained from Ningbo Zycalloy Co., Ltd., Ningbo, China. The chemical composition of these copper alloys is listed in Table 1. A 42CrMo steel plate was used as the backboard and its chemical composition is also shown in Table 1.

The lead–bronze plates (2 mm in thickness) and 42CrMo steel plates (8 mm in thickness) were abraded using 400# sandpaper to obtain a surface roughness (Ra) of c.a. 0.6 μm. Both plates were immersed in 10% NaOH solution at 80 °C for 10 min to remove the residual grease on the surface. Next, they were immersed in 10% HCl aqueous solution for about 10 min to remove the outer oxide layer. Finally, all metal plates were rinsed with deionized water and alcohol, respectively, and then dried for future use. A vacuum diffusion furnace (ZTF2-50-1, Jiangsu Bo Lian Shuo Welding Technology Co., Ltd., Xi’an, China) was used to prepare bimetallic alloy by vacuum welding technology. During this vacuum welding process, two types of metal plates were put into the furnace with the bronze plate on the bottom and the steel plate on the top. A pressure of 6 MPa was loaded on the two plates, as shown in Figure 1a. When the vacuum reached 10^−3^ Pa, the furnace began to heat to 680 °C with a heating rate of 5 °C/min, and this temperature was maintained for 60 min. After that, the furnace was cooled to room temperature by natural cooling. Before structural characterization and performance testing, all the bronze, brass and bimetallic materials were cut into small pieces of φ25 mm × 10 mm, as shown in Figure 1b. These pieces were further abraded and cleaned to obtain a uniform mirror surface with a roughness of less than 0.1 μm. 

### 2.2. Characterization

The microstructure of the specimens was observed by an optical microscope (Leica DMI 3000M, Leica, Weztlar, Germany) and a scanning electron microscope (SEM) with Energy Dispersive X-ray Spectroscopy (EDX). An EM-30AX Table-Top scanning electron microscopy (SEM) from COXEM Co. Ltd., Daejeon, Korea, equipped with an Oxford Xplore Compact 30 energy dispersive X-ray (EDX) spectrometer was operated at an electron accelerating voltage of 20 KV in secondary electron (SE) scanning mode. The work distance was 9 mm for EDX detecting. Before SEM analysis, the samples were mounted on an aluminum stub with carbon adhesive tape. Before the microstructure characterization, the top surface of samples was etched by acid solution. Before the examination of cross-sections of the bimetallic specimen, the cross-sectional samples were mounted with resin, well-polished and etched by acid solution. A Vickers microhardness tester was used to measure the hardness of the specimens under 0.098 N load with 12 s holding time. Ten points were measured on each sample to obtain the average hardness. The tribological properties of the specimens were examined by a pin-on-disc tribometer. The tests were conducted under the dry sliding test condition, with a sliding speed of 1000 revolutions/min (0.314 m/s). Two types of counterface balls with 6 mm in diameter were used in the tribological tests, of which one was made from ball bearing steel (GCr15 steel: C 1.0% and Cr 1.5% with HRC = 62) and another was cemented carbide (WC 94% and Co 6% with HRC = 92). The normal load was 1 N, i.e., 500 MPa contact pressure calculated by the Hertzian contact theory. The sliding tests were conducted for 20,000 revolutions (377 m). The wear tracks and worn counterface balls were investigated by a confocal microscopy (Leica DCM 3D, Leica, Weztlar, Germany) and SEM with EDX. The wear rate was calculated by the following equation:W = V/(F·S)
where W is wear rate (μm^3 ^N^−1 ^m^−1^); V is the wear loss volume (μm^3^); F is the normal load (N); and S is the sliding distance (m).

## 3. Results and Discussion

### 3.1. Microstructure

Figure 2 shows the SEM micrographs and the corresponding EDX spectra from the various microstructure regions of the three types of materials. The morphology and EDX analysis revealed that the lead–bronze was composed of α phase and soft Pb phase (Figure 2a,d) [31]. From the SEM image of Mn/Si–brass (Figure 2b) and the corresponding EDX spectra (Figure 2e), Mn_5_Si_3_ hard particles were evenly distributed into the matrix of α Phase and β phase [32]. The grain size of the Pb phase in the bimetallic sample (Figure 2c,f) was much larger than those in lead–bronze, which resulted from the grain growth during the heating process in vacuum diffusion welding. To further characterize the microstructure of bimetallic material, the SEM image of the cross-sectional morphology of the bimetallic specimen was shown in Figure 3a, together with the EDX line scanning spectrum from the bronze layer to steel layer demonstrated in Figure 3b. The two metals were bonded very closely and formed a continuous interface from the observation of SEM (Figure 3a). In addition, a diffusion layer was found at the interface of the two metals based on the EDX analysis. The diffusion layer thickness was roughly 10 μm with the main chemical composition of Cu and Fe. The diffusion layer is a chemical composition transition layer: the elements in the bronze diffused into the steel, so the copper content decreases gradually, and the steel content increases accordingly. When the copper disappears completely from the EDX spectrum (Figure 3b), the diffusion layer ends and reaches the steel matrix.

### 3.2. Microhardness

The average hardness results for three samples were demonstrated in Figure 4a. The brass possessed the highest hardness of about 323 Hv, followed by the lead–bronze alloy of 167 Hv, while bimetallic material had the lowest hardness of 130 Hv. Figure 4b shows the hardness test indents on various areas of the cross-sectional bimetallic sample. Apparently, the hardness increased gradually from the lead–bronze alloy bimetallic interface to the steel substrate, of which the hardness of steel bearing plate reached about 604 Hv. The variation tendency of hardness from the side of the bronze to the side of the steel showed that no hard and brittle interface layer was formed, different from the results reported in previous literature [33]. Thus, this diffusion layer could avoid the possible interface brittle rupture during the application process.

### 3.3. Tribological Properties

The coefficient of friction (COF) curves of the three samples are shown in Figure 5a–c, under the test conditions of 1 N normal load against GCr15 steel. The morphology of the counterface steel balls after the tests observed by optical microscope is also shown in Figure 5d–f. After a running-in stage, the COFs of the lead–bronze (Figure 5a) increased gradually from 0.7 to about 0.85 at the end of the sliding distance. Besides, this curve fluctuated violently and presented a stick-slip behavior during the sliding distance. While, for the Mn/Si–brass sample, after a running-in stage, its COF (Figure 5b) was about 0.3, then COF increased to about 0.5 in the first 200 m. In the next 30 meters’ sliding distance, the COFs rapidly increased to about 1.2. From then on, the COF curve fluctuated violently until the end of the friction test. Unlike the above two samples, the COFs of the bimetallic sample reached to about 0.8 after a short running-in stage and the COF curve was very smooth for the whole testing distance (Figure 5c).

To reveal the wear mechanism of these materials, SEM micrographs of the corresponding wear tracks are shown in Figure 6a–c, together with the EDX spectra detected from the labeled areas. In addition, the 3D images and 2D profiles of the corresponding wear tracks are also demonstrated in Figure 6d–f. From Figure 5d, it can be seen that the counterface GCr15 balls seriously abraded the lead–bronze, while in the wear track of the lead–bronze sample (Figure 6a,d) large amounts of plastic deformation and adhesive wear can be observed. Certain amount of wear debris consisting of Cu, Pb, Sn, Ni and O adhered to the wear track, which was further compressed during the tribological test. However, due to the low strength of the lead–bronze, large plastic deformation and ploughing occurred during the sliding process and the adhered layer presented a discontinuous top layer on the wear track. The sliding of this layer against the GCr15 steel and the occurrence of adhesive wear caused the stick-slip behavior of the COF curve (Figure 5a). Meanwhile, some of the wear debris could serve as the ‘third body’, which leads to the obvious abrasive wear in the wear track.

For the Mn/Si–brass sample (Figure 6b,e), little plastic deformation can be observed. This could be a benefit of the relatively higher hardness of the brass than the other samples. However, the wear track of the brass sample also showed obvious abrasive wear and hard particle embedding. The EDX spectrum indicates that the particles were mainly composed of Cu, Zn, Mn and Si. Thus, it can be postulated that severe abrasive wear can occur when the hard precipitations of Mn_5_Si_3_ accumulate at the sliding interface. This can also explain the gradually increasing behavior of the COFs for this sample, as shown in Figure 5b, and the abrasive wear of the corresponding GCr15 ball (Figure 5e).

The surface of the bimetallic sample was only polished slightly with a smooth and dense layer observed on the wear tracks (Figure 6c,f). The results of the EDX detected from the specific area (Figure 6c) show that the content of Fe and O elements increased significantly. Fe was mainly from the wear debris detached from the steel ball, and the wear debris with Cu, Pb, Fe and other minor elements were compressed and hardened to form a stable top layer. This top layer was more likely oxidized during the sliding processes. Compared with the discontinuous top layer formed on the lead–bronze alloy, the stable layer on the surface of the bimetallic structure was attributed to the supporting effect of the steel substrate. However, the corresponding counter ball showed the largest wear loss among the three samples (Figure 5f). From the 3D images and 2D profiles of the wear tracks (Figure 6d–f) for the three samples, the bimetallic sample showed the lowest wear rate and thus highest wear resistance, followed by lead–bronze and Mn/Si–brass, respectively.

Next, the tribological properties of three samples against the WC-based balls, which had higher hardness and more inert chemical properties, were investigated. The COF curves for these samples against WC balls under 1 N are given in Figure 7a–c and the optical microscopy images of the worn balls the after tests are shown in Figure 7d–f. In general, after wear tests, very little wear could be observed on the WC balls corresponding to each sample. For the lead–bronze sample, the COF value was stable at about 0.4 and the COF curve presented a stick-slip behavior. For the Mn/Si–brass sample (Figure 7b), the COF curve displayed a smooth state with COF of about 0.2 during the first 70 m. In the subsequent 100 m sliding distance, the COFs increased rapidly to about 0.8, after that, the COF curve fluctuated drastically around the value of 0.8 until the end of the test. The bimetallic sample displayed a low COF of about 0.2 at the first 100m sliding distance (Figure 7c). Further increasing the sliding distance led to the increase in COF, which reached about 0.9 at 300 m, and then COF was kept almost unchanged for the rest of the sliding tests.

Figure 8a–c are the SEM micrographs of the wear tracks against the WC balls under the 1 N load, with the corresponding EDX spectra detected from the labeled areas. Figure 8d–f demonstrate the 3D images and 2D profiles of the corresponding wear tracks. For the lead–bronze sample, large plastic deformation, ploughing and abrasive wear can be observed (Figure 8a,d). In comparison, the Mn/Si–brass sample showed higher wear but less plastic deformation (Figure 8b,e). However, for the bimetallic sample (Figure 8c,f), the surface was only slightly polished and very little wear was observed. A compressed dense top layer formed in the wear track.

Figure 9 summarizes the wear rate of the three samples against GCr15 and WC balls, respectively. When paired with the ball bearing steel, the wear rates of the lead–bronze and Mn/Si–brass were 13 and 54 times higher than that of the bimetal composite. When paired with bearing steel, the wear rates of the two materials were 13 and 54 times higher than the bimetallic composite, respectively. Although the Mn/Si–brass material possessed the highest hardness, it showed the highest wear rate. The hard precipitation that detached from the sample served as the “third body”, which accelerated the abrasive wear and exacerbated the wear resistance [34]. For the lead–bronze sample, it showed better wear resistance than the Mn/Si–brass alloy, which was associated with the formation of an oxidized lead-rich layer covering the wear track [35]. However, the wear debris containing Cu, Pb and Fe was more likely to be hardened at the interface. This hardened debris embedded in the sliding interface would give rise to severe adhesive wear and abrasive wear, as shown in Figure 6 and Figure 8. Of three samples, the bimetallic sample presented the highest wear resistance against both counterface balls. The reason is that the steel layer served as a bearing layer to decrease the plastic deformation of the bronze layer. Furthermore, the lead substance can also accelerate the formation of a dense hardened layer at the interfaces to avoid subsequent wear of the bronze surface. However, it should also be noted that this hardened layer causes severe scuffing on the steel balls, as shown in Figure 5f. Therefore, lead–bronze/steel bimetal structured material is better matched with hard counterface material, such as cemented carbide.

## 4. Conclusions

(1)The lead–bronze/steel bimetallic structure was produced by a vacuum diffusion welding technology. The lead–bronze layer was tightly bonded with steel and formed a continuous diffusion layer.(2)The tribological properties of three samples, i.e., lead–bronze, Mn/Si brass and bimetallic materials were tested. Under a dry sliding condition and high load (~500 MPa), large amounts of plastic deformation occurred for the lead–bronze, which was due to its insufficient mechanical strength and the failure to form a lubricating film by the soft lead phase. The wear debris with lead was easy to be hardened and accelerated the adhesive wear and abrasive wear. This situation was more serious when the counterface material was a relatively soft metal (GCr15).(3)Although the lead-free Mn/Si–brass alloy had higher hardness, its wear performance was worse than the lead–bronze under the dry sliding condition, which was due to the severe abrasive wear derived from hard precipitations in the alloy.(4)The lead–bronze/steel bimetallic structure can minimize the use of lead in the copper alloy. Meanwhile, this bimetallic structure achieved a better wear resistance than the other two alloys. The steel layer can serve as a bearing layer to decrease the plastic deformation of the bronze layer and accelerate the formation of a dense hardened layer at the interfaces to avoid subsequent wear of the bronze surface. Nevertheless, this hardened layer caused severe scuffing on the counterface materials.(5)Therefore, lead–bronze/steel-structured material had the best wear resistance among the three materials. It is recommended to be applied in the key friction pairs of Axial piston pumps matched with hard counterface material, such as cemented carbide.

## Figures and Tables

**Figure 1 materials-15-01588-f001:**
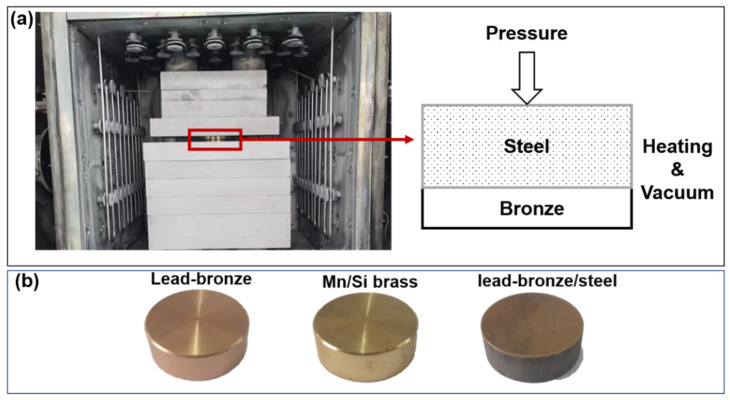
(**a**) photograph and schematic diagram of the vacuum diffusion welding layout (**b**) photographs of the three types of samples.

**Figure 2 materials-15-01588-f002:**
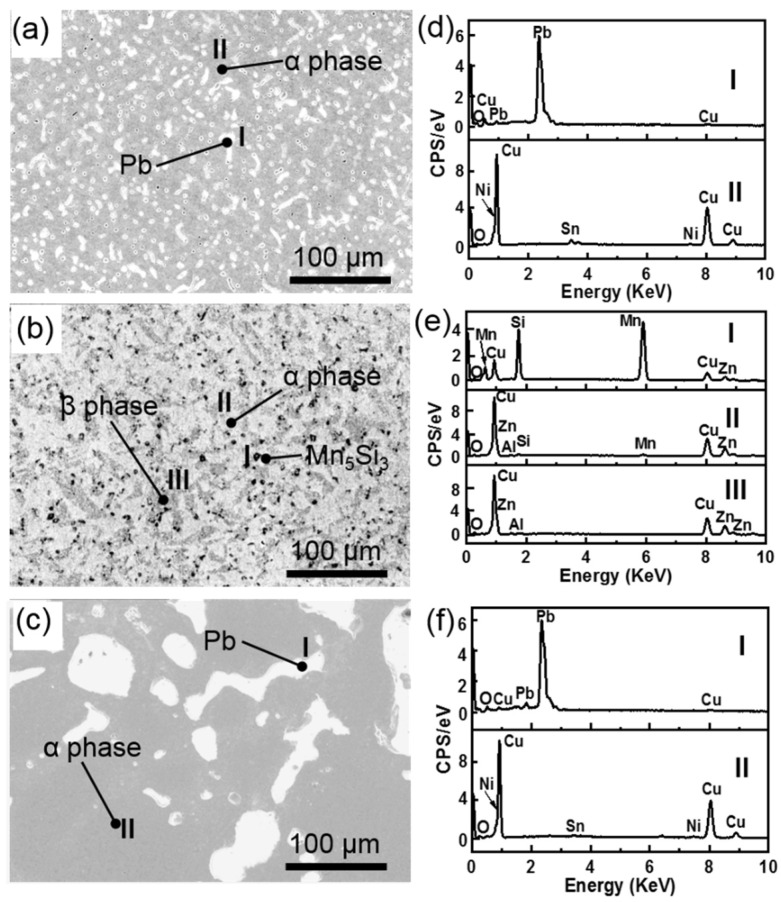
SE SEM micrographs of the (**a**) lead-bronze, (**b**) brass and (**c**) bronze layer of bimetallic sample. (**d**–**f**) are the corresponding EDX spectra detected from the marked points on (**a**–**c**), respectively.

**Figure 3 materials-15-01588-f003:**
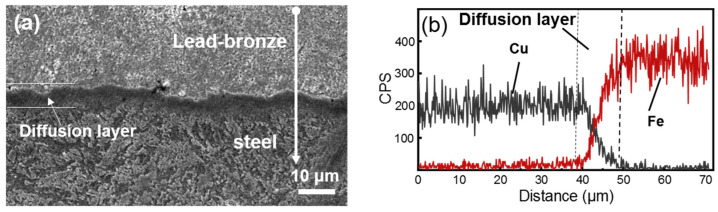
(**a**) SE SEM micrographs of the cross-section of the bronze–steel bimetallic specimen; (**b**) Line scanning EDX spectrum from the bronze layer to steel layer.

**Figure 4 materials-15-01588-f004:**
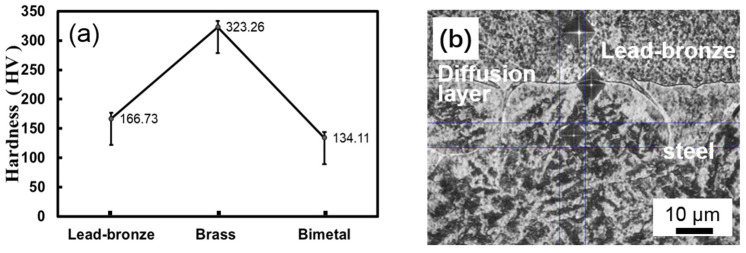
(**a**) Diagram of the hardness of the three materials and (**b**) Optical microscopy micrograph of the indents on the various areas of the cross-sectional bimetal sample.

**Figure 5 materials-15-01588-f005:**
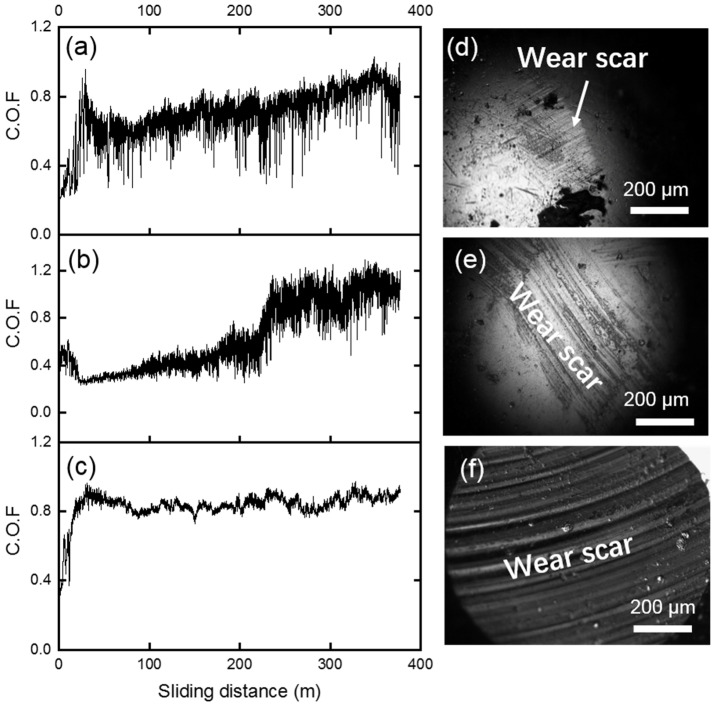
Coefficient of friction (COF) curves of (**a**) bronze, (**b**) brass, and (**c**) bimetallic specimens under 1 N normal load against GCr15 steel spheres. (**d**–**f**) The optical microscope images of the corresponding worn steel balls after the tests.

**Figure 6 materials-15-01588-f006:**
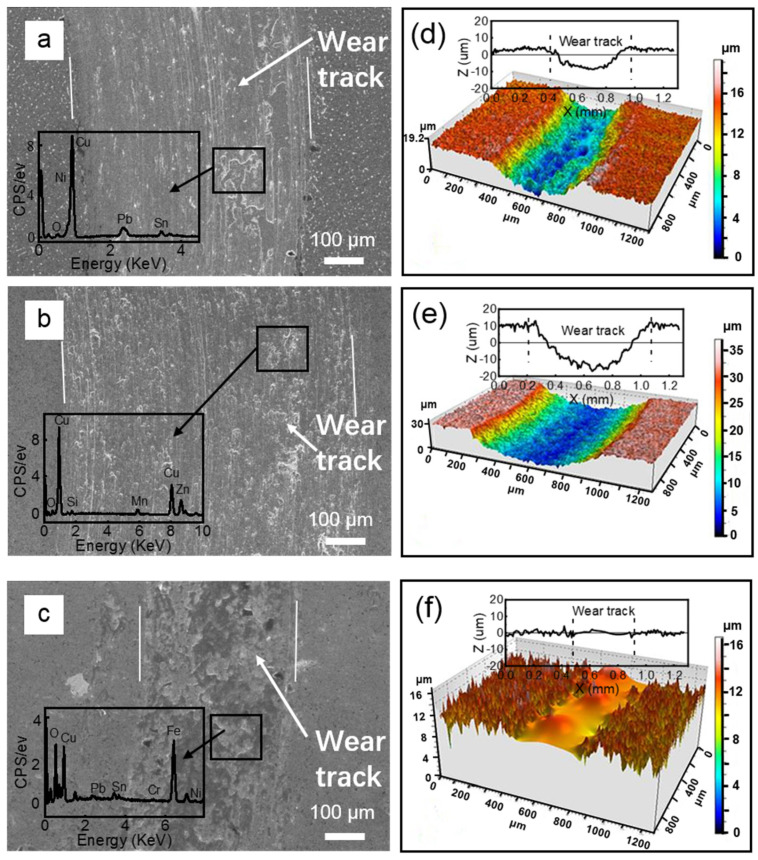
(**a**–**c**) SE SEM micrographs of the wear tracks of (**a**) bronze, (**b**) brass and (**c**) bimetallic samples. The inserts in (**a**–**c**) are the corresponding EDX spectra detected from the labeled areas. (**d**–**f**) 3D images and 2D profiles of the corresponding wear tracks of (**d**) bronze, (**e**) brass and (**f**) bimetallic samples against the GCr15 under 1 N.

**Figure 7 materials-15-01588-f007:**
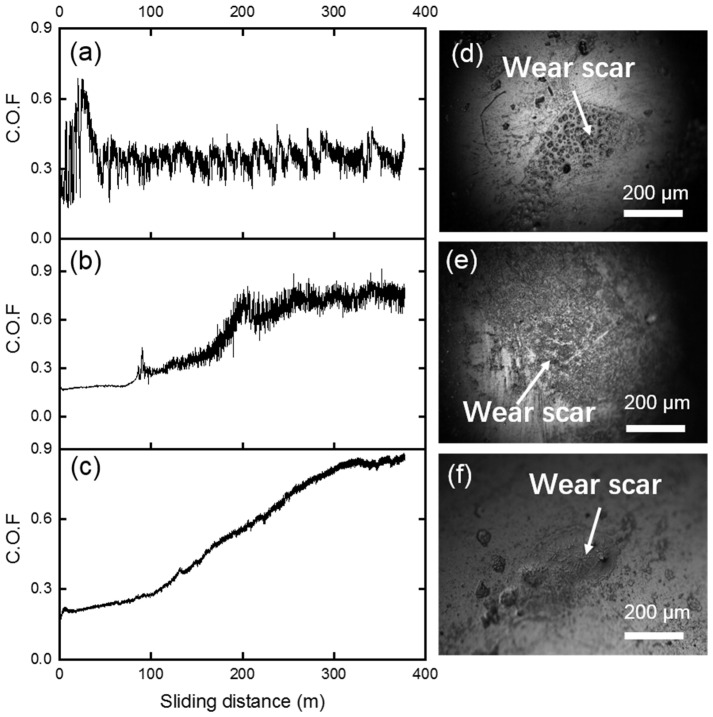
Coefficient of friction (COF) curves of (**a**) bronze, (**b**) brass and (**c**) bimetallic specimens under 1 N normal load against WC spheres. (**d**–**f**) The optical microscope images of the corresponding worn WC balls after the tests.

**Figure 8 materials-15-01588-f008:**
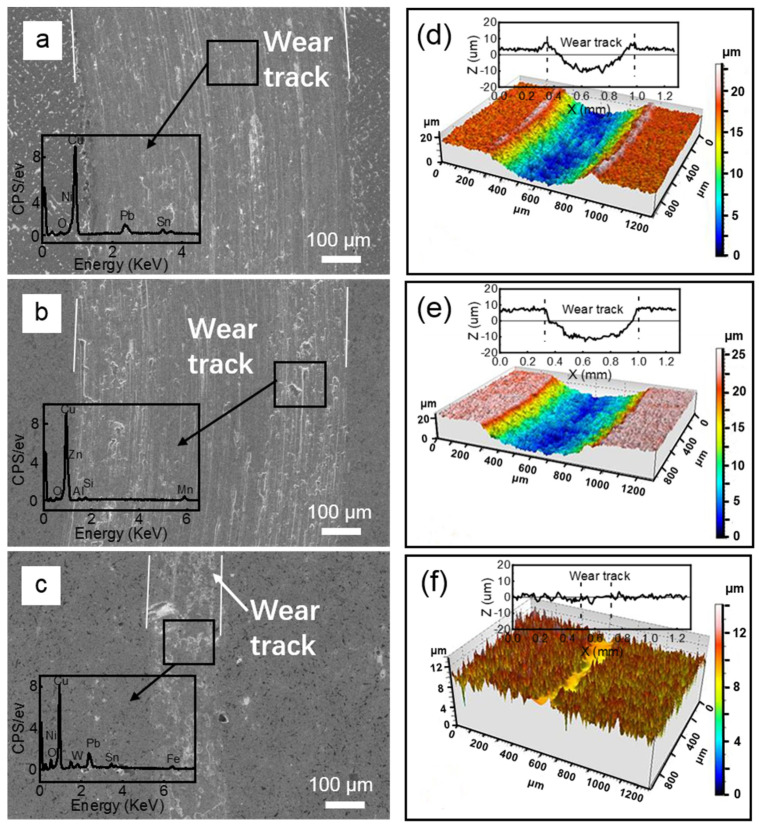
(**a**–**c**) SE SEM micrographs of the wear tracks of (**a**) bronze, (**b**) brass and (**c**) bimetallic samples. The inserts in (**a**–**c**) are the corresponding EDX spectra detected from the labeled areas. (**d**–**f**) 3D images and 2D profiles of the corresponding wear tracks of (**d**) bronze, (**e**) brass and (**f**) bimetallic samples against the WC balls under 1 N.

**Figure 9 materials-15-01588-f009:**
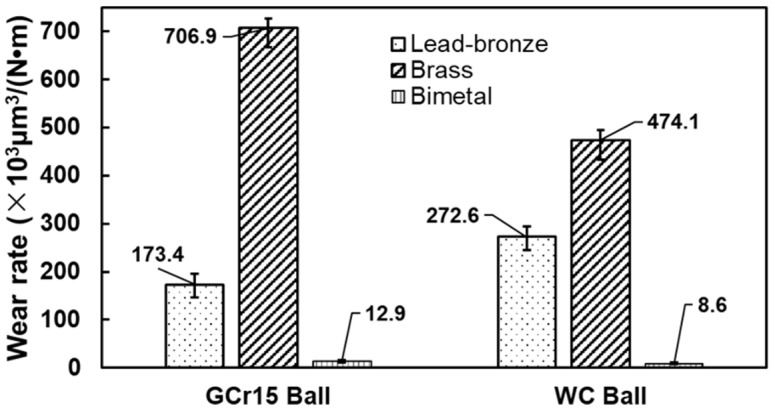
Diagraph of the wear rate of the three samples against GCr15 and WC balls.

**Table 1 materials-15-01588-t001:** The chemical composition of the copper alloys and 42CrMo steel.

(wt %)	Cu	Zn	Pb	Mn	Si	Al	Fe	Sn	Ni
Lead–bronze	78.54	<0.03	15.35	-	-	-	-	4.75	1.33
Mn/Si–brass	58.53	35.65	<0.5	2.79	0.77	1.26	<0.35	<0.3	<0.25
**(wt %)**	**C**	**Si**	**Mn**	**P**	**S**	**Cr**	**Mo**	**Fe**	
42CrMo steel	0.40	0.27	0.83	<0.035	<0.04	0.10	0.20	balanced	

## Data Availability

Not applicable.

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
