# Peer review of "Microstructure, Microhardness and Tribological Properties of Bronze–Steel Bimetallic Composite Produced by Vacuum Diffusion Welding"

_materials, 2022, doi:10.3390/ma15041588_

Round 1
Reviewer 1 Report
This manuscript entitled, “Tribological Properties of Bronze-steel Bimetal Composite Produced by Vacuum Diffusion Welding” is interesting and falls under the scope of reputed Materials journal. I really appreciate this objective approach. While reading, I had remarks and comments that may be helpful in preparing the revised version of the article. Overall, I think that this manuscript could be considered for publication if the Authors will be able to take into account the following major revisions:
- After each terminology firstly appear in this paper, it necessary to explain. Add nomenclature at the end.
- Add some qualitative and quantitative findings in abstract.
- Reference style is not as per the journal format. Check it again.
- Introduction section is not written properly. Justify the selection of material and the selected approach with proper justification, use and applications.
- I didn’t observe any previous literature work in introduction section. Authors must explain the research gap by giving proper explanation of past work done by researchers.
- Add relevant citations in introduction section. Preferably more from materials, mdpi.
- Include the photograph of the used machining set-up and samples used in the study
- Line 30 – sentence without support. Add relevant reference.
- Line 115-116: Justify the sentence. What is the reason for, “The Cu content gradually decreases but Fe content increases from the bronze layer to the steel substrate”
- What is the holding time for Microhardness measurement? Mention it
- Add citations in results and discussion section which should justify your sentences. I didn’t observe any citation in this section.
- Improve the quality of Figure 6 and 8. Scale bar is not readable
- Justify: Bimetallic structure achieved a better wear resistance than the other two alloys.
- Rewrite the conclusions in order to highlight the limitations, improvements of present work and future scope of study. Write the conclusion section point wise.
Reviewer 2 Report
This manuscript is an investigation on microstrcture, microhardness and tribological properties of bimetallic composite bronze-steel produced by vacuum diffusion welding. The manuscript is within the scope of this journal. However, following points must be taken into consideration before consideration for publication:
- Title must be changed as suggested 'investigation on microstrcture, microhardness and tribological properties of bimetallic composite bronze-steel produced by vacuum diffusion welding' or similar, as it discusses about all three aspects.
- List References must be enhanced considering following references: https://www.sciencedirect.com/science/article/pii/S1526612521004631; https://link.springer.com/article/10.1007/s12540-020-00759-w
- Discussion part must be improved considering scientific discussions and comparing obtained results with published literature.
- English proof reading is recommended to meet international standard of this journal.
Reviewer 3 Report
The subject matter is very interesting, important, and has a special value considering practical applications. References are adequate considering the discussion in the paper. The paper is not clearly presented and well organized. There are still some things that could be improved, and a few questions that have to be answered before publication. Therefore, I suggest a mandatory revision of the following points to increase the quality of the paper:
1. There is no information about the EDS and SEM methods. The authors should provide the names of all devices as well as parameters.
2. The authors have described the individual phases in Fig. 2. The description was made on the basis of EDS method. They should put in the results of the study on the chemical composition.
3. In addition, information should be provided in what contrast were the photos taken, ie. SE or BSE?
4. On the SEM images showing the worn surfaces, the places described in the manuscript should be marked with arrows and described.
5. Have the authors tried to determine the thickness of the tribofilm?
Recommendation:
This manuscript in the presented form is not acceptable for publication in the Materials. The major revision is necessary.
Round 2
Reviewer 1 Report
Authors has addressed all my comments from previous round. Manuscript is improved significantly. In my opinion, it can now be accepted in present form.
Reviewer 3 Report
The authors have taken into account the reviewer's comments and they have made corrections in this article. They have responded to all comments of the reviewer. The authors have performed significant positive changes to the manuscript. The work is clear and very interesting.
Recommendation:
This manuscript in the presented form is acceptable for publication in the Materials.